# Healthy Food Policies Documented in University Food Service Contracts

**DOI:** 10.3390/ijerph20166617

**Published:** 2023-08-21

**Authors:** Stacy M. Fandetti, Alicia Anne Dahl, Caitlan Webster, Morium Barakat Bably, Maren J. Coffman, Elizabeth F. Racine

**Affiliations:** 1Department of Public Health Sciences, University of North Carolina at Charlotte, 9201 University City Boulevard, Charlotte, NC 28223, USA; adahl3@charlotte.edu (A.A.D.); cwebst13@charlotte.edu (C.W.); moriumbably@gmail.com (M.B.B.); 2School of Nursing, University of North Carolina at Charlotte, 9201 University City Boulevard, Charlotte, NC 28223, USA; m.coffman@charlotte.edu; 3Texas A&M AgriLife Research, Texas A&M University, 1380 A and M Circle, El Paso, TX 79927, USA; beth.racine@ag.tamu.edu

**Keywords:** university food environment, emerging adults, food service contracts, food environment policy, university students

## Abstract

In the United States, there is an opportunity to improve the nutritional health of university students through the campus food environment. This project used a content analysis approach to investigate whether healthy food standards and policies were incorporated into the contract agreements between North Carolina (NC) public universities and their food service management companies. Food service contracts were collected from 14 NC public universities using food service management companies on campus. Each contract was evaluated using the 35-item North Carolina Food Service Policy Guidelines Assessment to examine four elements of the campus food environment: Beverages, Packaged Snacks, Prepared Foods, and Other (e.g., strategic placement of healthier food). Five university food service contracts incorporated no North Carolina Food Service Policy Guidelines, three university contracts included one to five guidelines, and six university contracts included six to nine guidelines. Altogether, 13 of the 35 guidelines were incorporated into at least one university food service contract. This project presents a cost and time-effective assessment method for determining if evidence-based nutrition guidelines have been included in university food service contracts. This approach and findings may lead to contract revisions to improve the campus food environment and, subsequently, the nutritional health of college populations.

## 1. Introduction

The food environment influences a person’s food choice and subsequent nutritional outcomes. The external food environment has been described as consisting of food availability, prices, vendor and product properties, marketing and regulation, and policies [1]. Additionally, a personal food environment describes the individual-level influences on food choices, such as accessibility, affordability, convenience, and desirability [1]. Globally, other factors of consideration in food choice are food safety, social forces, gender dynamics, and stability [2]. These external and personal factors interplay within the context of a food system and become drivers of food choice across diverse settings [2].

Organizational food environments, recognized by some theoretical models as a subset of the broader external food environment, are those available to defined groups and include companies, healthcare facilities, and universities [3,4]. Examining the relationship of organizational food environment components may lead to a better understanding of this environment and more effective interventions to improve them [3,4]. In particular, the university food environment includes unique elements such as being composed of emerging adults (ages 18 to 29) in which the organization provides some degree of responsibility for basic needs such as housing and food [5,6].

Emerging adulthood is a developmental life stage, and the transition of behavioral norms from the family food environment to adulthood is complex [6]. In 2021, 38% of emerging adults in the United States (US) were enrolled in college [7], with current estimates projecting an increase of 8% by 2030, totaling 17.5 million students [8]. Thus, the college environment is ideal for exploring food choices among this age group. Understanding the influence of the college food environment on food choice has the potential to impact a large proportion of the emerging adult population’s long-term health [9].

Many studies suggest that behavioral lifestyle factors, including poor diet quality and eating behavior, are the primary causes of increasing obesity prevalence and excessive weight gain among emerging adults [10,11,12]. Previous research suggests college students have low intakes of fruits and vegetables [13], high intakes of fats [13,14], and high intakes of added sugars [14]. Most universities offer meal plans and food retailers on campus to provide students the convenience of staying on-site for meals; however, there is often limited support for healthy eating [15,16,17,18]. College students are more likely to be influenced by unhealthy food environments due to limited time for grocery shopping, inadequate cooking skills, and budget constraints [19,20,21,22]. Other factors influencing food choice include proximity to campus food outlets, dietary preferences, peer pressure, and environmental sustainability considerations [19,20,21,22,23,24].

In the US, many universities outsource their dining operations to a food service management company [18]. University food service is one of the largest sectors of the food service industry, grossing over $18 billion in 2018 [25]. Many contracts have an initial term of 5–10 years, with an option for renewal. Within these contracts, the hired food contractor must specific the scope of management. Still, it depends on the university department that oversees contracted services (e.g., housekeeping, parking, and food) to monitor the contract terms [26]. Contractors are responsible for serving food in various university settings, such as student dining halls, quick service cafes, and catering for university events. A university’s decision to outsource or self-operate dining services is multifaceted and involves evaluating many factors, including financial performance, quality standards, and capital considerations [27].

In recent years, there has been a growing interest in examining food choices and dietary habits among college students as well as the healthfulness of the university food environment [28]. Much of the research examines predictors of healthy food purchasing [13,18,19,29,30,31,32,33,34] and perceptions of the campus environment [19,35,36,37,38]. Additionally, some research has investigated the quality of food and beverages available to college students [5,16,17,39,40,41,42,43]. However, most articles do not specify if the included universities self-operate or outsource food service operations. In addition to the research described above, organizations such as Menus of Change University Research Collaborative and Partnership for a Healthier America have engaged in university-based nutrition assessments and interventions [44,45].

Implementing policies focused on enhancing the food environment can have a broader, more equitable, and more sustainable impact than those focused on individual behavior change [46]. For instance, a study of 4870 middle school students across 40 states found that evidence-based school meal standards were associated with more favorable weight status, indicating the potential for meal standards to address obesity rates [47]. Currently, no federal policies aim to improve the food environment in tertiary education [18]. However, national guidelines exist that universities may follow, such as the US Centers for Disease Control and Prevention (CDC) Food Service Guidelines for Federal Facilities [48]. The CDC Food Service Guidelines provide a set of standards to ensure the availability of healthy food and beverages, making it easier for consumers to make healthy choices [48]. Additionally, individual institutions may have their own food environment policies, but there is no known reporting agency or enforcement mechanism.

Publicly available university food service contracts can provide valuable information about university healthy food policies or nutritional standards. The current literature does not address the university and food service management company relationship and how this relationship influences students’ healthy food options. Thus, reviewing the contracts between food service providers and universities is a helpful starting point to determine if meal standards were considered during negotiations. This project aims to determine if healthy food policies were incorporated into the contract agreements between NC public universities and their food service contractors.

## 2. Materials and Methods

### 2.1. Setting and Sample

The University of North Carolina system consists of 16 public universities [49]. Among the 16 universities, 2 self-operate their dining services, and 14 universities use food service management companies for nutrition on campus. Despite being a part of an extensive university system, each school has its own food service contract: seven use Aramark, three use Compass USA-Chartwells, three use Sodexo, and one uses Thompson Hospitality. This project did not involve human subject research; therefore, it did not require review by the University of North Carolina at Charlotte Institutional Review Board.

### 2.2. Variable Measurement

To evaluate the food service contracts of 14 public universities, we used the North Carolina Food Service Policy Guidelines Assessment (see Appendix A). This assessment was developed by the NC Division of Public Health, Community and Clinical Connections for Prevention and Health Branch (CCCPH), in 2017 for organizations serving or selling food in cafeterias, vending machines, concession stands/snack bars, meetings/conferences, and other institutional settings [50]. It can be used as a baseline and to measure progress as changes have been implemented. The assessment includes criteria from the CDC Food Service Guidelines (CDC-FSG) for Federal Facilities document [48], two guidelines related to NC-specific campaigns, and one sustainability guideline. The CDC-FSG, based on the 2015–2020 Dietary Guidelines for Americans, was developed by experts from various federal agencies [48]. Additionally, the CDC-FSG is intended for developing contracts and permits for food service delivery, so we deemed this an appropriate measure for examining the contracts between universities and food service companies [48].

The North Carolina Food Service Policy Guidelines Assessment comprises 35 policy guidelines related to specific food items that fall within four categories: Beverages (4), Packaged Snacks (7), Prepared Foods (16), and Other (8). Beverages include water, milk, 100% juice, soft drinks, energy drinks, teas, and coffees [48]. Packaged snacks include processed foods packaged in small portions or individual servings with a relatively long shelf-life, such as granola bars, chips, and nuts [48]. Prepared foods are fresh, cleaned, cooked, or assembled and served as ‘ready to eat’ [48]. Prepared foods include those made and served on-site or those prepared at a central kitchen and packaged and distributed to other locations. These foods have a relatively limited shelf-life and can be sold in any food service venue, including hot entrees, side dishes, and sandwiches [48]. The Other category includes behavioral design components such as product placement, layout, and price incentives.

### 2.3. Project Design and Data Collection

This project was a content analysis of food service contracts. To be included, public NC universities needed to utilize a food service management company for nutrition on campus. Universities that self-operated dining services were excluded from this project because they did not have food service contracts to review.

In the US, the Freedom of Information Act allows the public to request public records from governmental bodies, such as public universities [51,52]. Accordingly, since all the universities in our analysis were public, the food service contracts were obtained through each campus’s legal office via a public records request. The public records requests were made in 2019, and contracts were emailed to the research team within 2–8 months of the respective request. Next, a trained research assistant extracted data from each contract, then a second research assistant reviewed and verified the data. Any discrepancies between the researchers were discussed among the team until a 100% agreement was reached. An Excel spreadsheet was created for each contract in which the following variables were recorded: university name, food service management company, contract date, relevant page numbers, the type of setting the contractor used to sell or serve food, and category of food sold by the contractor. Additionally, the contracts were examined for each of the 35 guidelines from the North Carolina Food Service Policy Guidelines Assessment; if a guideline was present, it was indicated with a ‘1’ and absence with a ‘0’. Moreover, if a guideline was identified in a contract, the associated contract language was copied, and the page number was included for reference.

Research assistants searched university websites to identify each university’s food service contractor director (e.g., Chartwells food service director) and the university staff member responsible for the food service contract. In early 2020, the principal investigator emailed the food service contracts and Excel spreadsheets with contract findings to the identified food service leadership at each of the 14 public universities to corroborate our observations. Campus food service leadership was asked to verify the authors’ findings and provide feedback. Specifically, we asked university leadership to indicate if nutritional guidelines were followed on campus as a ‘policy’, ‘non-contractual practice’, or neither. University staff were asked to email their assessment review to our research team. To encourage the review and return of the findings, research staff emailed the university representatives twice (spring 2020 and spring 2021). For non-responsive universities, the Director of Food Service at the research team’s university contacted the non-responsive universities a third time in mid-2021.

### 2.4. Data Analysis

Universities were classified as having a policy if a North Carolina Food Service Policy Guideline was written into the food service contract (e.g., documentation of an item such as 100% fruit juice with no added sugar). Using Microsoft Excel, descriptive analyses were conducted to quantify the number of universities reporting that the guidelines were followed as a policy. Descriptive analyses were also used to calculate the mean and median contract length. Moreover, certain universities reported that they voluntarily followed guidelines as non-contractual practices; therefore, we quantified non-contractual practices as separate variables. We examined policies and non-contractual practices separately and looked at policies and non-contractual practices together for the universities that provided feedback.

## 3. Results

Of the 14 eligible NC public universities with food service contracts, we received 14 of the 14 requested. Each university contract highlighted that food service providers were responsible for selling or serving food in a combination of at least two campus settings (see Table 1). All 14 food service contracts included cafeterias/cafes and catering services, and only 2 included providing meals through university hospitals (e.g., patient meals). To demonstrate the variety between universities, one institution provided food across all five settings, while more institutions (*n* = 5) stated that the vendor served food across four locations: cafeterias/cafes, catered events, concession stands, snack bars and carts, and vending machines.

When investigating the presence of healthy food guidelines, 5 of the 14 university food service contracts had no nutritional food guidelines explicitly mentioned, three contracts between one and five guidelines, and six contracts between six and nine guidelines (see Table 2). The mean and median length of a North Carolina public university food service contract was 14 years, ranging from 7 to 24 years. Table 2 shows no relationship between the year the contract originated or the food service management provider and the number of policies incorporated into food service contracts.

In total, 13 of the 35 NC Food Service Policy Guidelines were incorporated into at least one university contract (see Table 3). Three of the four categories on the NC Food Service Policy Guidelines Assessment were found in the university food service contracts: Beverages, Prepared Foods, and Other. No universities included Packaged Snacks guidelines in their contracts. Additionally, none of the guidelines were incorporated into all university contracts, and 22 guidelines were not evident in any agreements.

Two of the four Beverage category guidelines were identified in the food service contracts. Six universities included ‘offer low-fat milk with no added sugar’, making it the most frequently documented Beverage guideline. Seven of the 16 Prepared Foods guidelines were incorporated into food service contracts by at least one school. The most commonly observed Prepared Foods guideline was ‘offer protein foods from plants such as legumes, nuts, seeds, and soy products’, of which eight universities included in their contracts. The following most frequent guideline in this category, identified in seven university contracts, was ‘provide calorie and nutrition information of standard menu items’. Four Other guidelines related to behavioral design aspects of the food environment were identified in NC public university contracts. Two universities included ‘use product innovation and inclusion of healthier options as default choices’, and two had ‘make changes to vendor contracts that support food service guidelines’.

### Feedback from Universities

Seven of the fourteen universities (50%) responded to our inquiry regarding the accuracy of the food service contract content analysis. There were no apparent differences between universities that did not respond and those that provided feedback, such as geographic location, campus size, or food service management provider. See Table 4 for a summary of the subsamples’ total policies and non-contractual practices. Based on the feedback, we found that although a contract may not mandate specific policy guidelines, many universities follow them as a non-contractual practice. Of the seven responses we received, two schools did not indicate any additional non-contractual practices beyond what we found in their contracts. The other five schools reported 2 to 14 non-contractual practices not included in their contracts. For instance, five universities stated they ‘offer seafood at least two times per week,’ and four said they ‘provide at least three non-fried vegetable options at all times.’ When examining the total number of policies written into contracts and non-contractual practices reported by university representatives, 22 of 35 guidelines were adopted at one or more universities.

## 4. Discussion

This project evaluated the frequency of NC Food Service Policy Guidelines incorporated in NC public university food service contracts. There are 16 public universities in NC: 14 schools utilized a food service management company for nutrition on campus, and two were self-managed. We determined that 13 of the 35 NC Food Service Policy Guidelines were incorporated into at least one university contract. The maximum number of policies adopted by a university was nine of 35.

The first key finding of the project is that most universities incorporated very few healthy food policies into their food service contracts. Further, the number of policies documented in food service contracts did not have a clear relationship with the contract origination date or food service management company. For instance, the two most recent contracts, from 2014 and 2015, did not include any healthy food policies, whereas a contract from over 20 years included eight policies. All university food service contracts predated the NC Food Service Policy Guidelines Assessment developed in 2017. As a result, the universities did not have access to the assessment when negotiating their initial food service contracts. Moreover, the CDC-FSG for Federal Facilities was released in 2011 and updated in 2017 [48]. Future dissemination efforts should promote the CDC-FSG or comparable guidelines (e.g., Partnership For A Healthier America guidelines [45]) for inclusion in new food service contracts or contract addendums. Future research could evaluate how university and food service management company leadership views the guidelines and the feasibility of adopting guidelines as policies in food service contracts.

Our analysis of food service contracts showed that food service providers individualize the services offered at each institution. This finding presents an opportunity to address the organization’s need to consider the nutritional health of the student population served through food service contracts. Additional research could investigate why some universities have more guidelines than others within the same university system.

The second key finding is that the most frequently identified guideline was ‘offer protein foods from plants such as legumes, nuts, seeds, and soy products,’ which was included in eight university contracts. This guideline may be a priority among some universities due to the increase in the number of students who are vegetarians or desire meat alternatives [53,54]. More students are adopting plant-based diets for health, environmental, and ethical reasons [55]. Moreover, recent college students have demonstrated increasing concern for the environment and the sustainability of their food, regardless of their dietary patterns [56].

The third key finding is that none of the university contracts included any Packaged Snack guidelines. This finding is important because packaged snacks are ubiquitous on college campuses in the US, particularly in convenience markets and vending machines. Notably, university vending machines have been found to have a high prevalence of nutritionally poor food items documented in the literature [38,41,42,43,57,58,59]. For example, according to a cross-sectional analysis from an urban university, 95% of the food offerings and 49% of the beverages in vending machines were unhealthy [43]. Evaluation of packaged snacks is vital because many young adults engage in unhealthy snacking. For instance, one study reported that one-third of young adults consumed more than six unhealthy snacks a day [12]. Seven of the contracts (50%) indicated that vending machines were included within the food service contractor’s responsibility scope. The other universities likely utilize a separate vending machine contractor to manage that component of their food environment, which may have influenced the lack of packaged snack policies. Further research to explore the contracts with vending machine operators is warranted and may help universities engage in conversations to prioritize and negotiate nutritionally healthy vending machine offerings.

Although food service contracts are easily accessible through public records requests, feedback from universities to validate the findings was more challenging to obtain. The input from university representatives (*n* = 7) provided rich insight into the non-contractual practices found in university food environments. It highlighted that food service contracts may only contain a partial picture of healthy food guidelines on campus. From our sub-analysis (*n* = 7), we learned that additional efforts aligned with healthy food guidelines (e.g., free access to chilled drinking water) are implemented outside of food service contracts. This finding presents an opportunity for universities to integrate more non-contractual practices as contractual policies, thus making them enforceable. Future work should incorporate developing strategies to improve the likelihood of obtaining university feedback to gain a thorough understanding of the university food environment. For instance, providing a stipend for participation or fostering more vital collaboration with the broader university system may improve response rates and the quality of feedback.

To our knowledge, this was the first project to examine if nutritional standards were incorporated into university food service contracts. The contracts between public universities and food service companies are public information; therefore, this relatively simple form of data collection can be easily replicated at other institutions. A strength of the project is that feedback from campus leadership was requested and included in the final analysis. This approach validated the authors’ findings and provided additional insight into the campus food environment not contained in the food service contracts. Moreover, the project used the NC Food Service Policy Guidelines Assessment, which is based on CDC recommendations for food service delivery. Finally, the project included food service contracts from all public NC universities that partnered with a food service management company, creating a representative content analysis for our state.

There are limitations to the project. First, the project team asked for the original food service contracts, not addendums or annulments. Contracts are updated periodically when food items or prices change; therefore, a policy may have been in an addendum and not reflected in our results. Second, the feedback stating that many guidelines were non-contractual practices rather than policies suggests that the food service contracts may only reflect some aspects of the campus food environment. Other factors of food service contracts may influence students’ health, such as dining operating hours, the price of meal plans, and types of on-campus restaurants. Future research should explore these aspects to develop additional insight into the food service management company and university relationship. Third, although we received feedback from 50% of the universities, having a higher response rate could have affected the results observed. Finally, this analysis focused on public universities in one state; therefore, the findings may not be generalizable to other universities.

## 5. Conclusions

University food environments, a type of organizational food environment, exert power and influence over students’ food choices, including controlling the food available for purchase. As public health nutrition practitioners identify strategies to mitigate obesity among the emerging adult population, focusing on improving the college food environment may be a practical approach. Without a centralized way of regulating the nutritional standards of college food environments, documenting policies in a food service contract may be a starting point for future evaluation and improvement efforts. Universities have an opportunity to keep health at the forefront of contract negotiations with food service providers. For instance, inviting public health nutritionists, students, and other campus stakeholders into the contract negotiation process, or providing a forum for submitting feedback, may lead to incorporating students’ nutritional health considerations into university food service contracts.

This paper examined one element of the campus food environment–food service contracts. A more detailed assessment by researchers and campus leadership can supplement the findings obtained from food service contracts. In addition to examining contracts, universities are encouraged to assess student preferences, the financial value of food, and environmental sustainability through surveys or focus groups. In our experience, we have found that it is valuable and imperative to involve campus and food service management company leadership to adopt evidence-based practices. This project presents an assessment method for engaging campus dining stakeholders in a conversation about implementing evidence-based nutrition guidelines as a priority in the decisions about campus food offerings.

## Figures and Tables

**Table 1 ijerph-20-06617-t001:** Types of Settings Food Service Contractors were Charged with Managing Within the University of North Carolina System.

University Food Service Contractor Setting Type	Number of NC Public University Contracts N = 14
Cafeterias/cafes	14
Catered events	14
Concession stands, snack bars, and carts	11
Vending machines	7
Hospital patient meals	2

**Table 2 ijerph-20-06617-t002:** University Food Service Contracts: Date of Origination and Number of Policies Present.

NC Public Universities with Food Service Contracts N = 14	Year Contract Originated	Number of Policies Present in Food Service Contracts
University 1	1998	0
University 2	1999	0
University 3	2001	8
University 4	2006	7
University 5	2006	0
University 6	2007	8
University 7	2008	3
University 8	2009	9
University 9	2010	8
University 10	2012	1
University 11	2013	8
University 12	2014	4
University 13	2014	0
University 14	2015	0

**Table 3 ijerph-20-06617-t003:** Policies Documented in Food Service Contracts of North Carolina Public Universities.

Category	North Carolina Food Service Policy Guidelines	Number of Policies Present in University Food Service Contracts N = 14
Beverages	1. Free access to chilled drinking water	0
	2. Low-fat milk with no added sugar	6
	3. 100% juice with no added sugar	4
	4. At least 50% of beverages ≤40 calories/8 fl oz	0
Packaged Snacks	5. All packaged snacks contain ≤200 mg of sodium	0
	6. All packaged snacks have zero grams of trans fat	0
	7. All packaged snacks at least one: 1st ingredient fruit/vegetable/dairy/protein; at least ¼ fruit/vegetable; at least 50% whole grain	0
	8. All packaged snacks contain ≤200 calories per package	0
	9. All packaged snacks contain <10% of total calories from saturated fat	0
	10. All packaged snacks contain ≤35% of weight from total sugars	0
	11. All vending machines are consistent with FDA Vending Machine Final Rule: Food Labeling/Calorie Labeling	0
Prepared Foods	12. At least three fruit options with no added sugars at all times	6
	13. At least three non-fried vegetable options at all times	2
	14. Seasonal fruits and vegetables	5
	15. Half of the total grains offered as “whole-grain rich” and the remaining grains enriched	0
	16. Variety of low-fat dairy products/alternatives such as milk, yogurt, cheese	5
	17. Variety of non-fried protein foods such as seafood, lean meats, and poultry, eggs, legumes, nuts, seeds, soy products	6
	18. Protein foods from plants such as legumes, nuts, seeds, and soy products	8
	19. Seafood at least two times per week	0
	20. 25% of desserts contain ≤200 calories	0
	21. All meals contain ≤800 mg sodium	0
	22. All entrees contain ≤600 mg sodium	0
	23. All side items contain ≤230 mg of sodium	0
	24. All foods free of partially hydrogenated oils	0
	25. Provide calorie and nutrition information of standard menu items	7
	26. Deep-fried options limited to no more than one choice per day	0
	27. At least 10% spent on products originating in North Carolina	2
Other	28. Organization signed up for North Carolina 10% campaign	0
	29. Strategic placement to foster the selection of healthier foods and beverages	0
	30. Product innovation and inclusion of healthier options as default choices	2
	31. Price incentives and marketing strategies used to highlight healthier food and beverage items	0
	32. Healthy portion sizes are promoted by optimizing the size of dishware and serving ware	0
	33. Displays, decorations, and signage used to highlight healthier choices	1
	34. Healthier options promoted via worksite wellness programs or other employee organizations	0
	35. Changes to vendor contracts that support food service guidelines	2

**Table 4 ijerph-20-06617-t004:** Universities That Responded to Our Inquiry (*n* = 7): Results from Their Feedback to Verify Policies and Non-Contractual Practices.

Category	North Carolina Food Service Policy Guidelines *	Policies Identified in University Contracts *n* = 7	Non-Contractual Practices Reported by University Representatives ** *n* = 7	Total Policies and Non-Contractual Practices for Sub-Sample *n* = 7
Beverages	Free access to chilled drinking water	0	3	3
	2. Low-fat milk with no added sugar	3	3	6
	3. 100% juice with no added sugar	2	2	4
Packaged Snacks	4. All packaged snacks have zero grams of trans fat	0	2	2
Prepared Foods	5. At least three fruit options with no added sugars at all times	3	3	6
	6. At least three non-fried vegetable options at all times	1	4	5
	7. Seasonal fruits and vegetables	3	3	6
	8. Half of the total grains offered as “whole-grain rich” and the remaining grains enriched	0	2	2
	9. Variety of low-fat dairy products/alternatives such as milk, yogurt, cheese	3	3	6
	10. Variety of non-fried protein foods such as seafood, lean meats, and poultry, eggs, legumes, nuts, seeds, soy products	3	3	6
	11. Protein foods from plants such as legumes, nuts, seeds, and soy products	3	2	5
	12. Seafood at least two times per week	0	5	5
	13. Provide calorie and nutrition information of standard menu items	3	2	5
	14. At least 10% spent on products originating in North Carolina	0	1	1
Other	15. Organization signed up for North Carolina 10% campaign	0	2	2
	16. Strategic placement to foster the selection of healthier foods and beverages	0	1	1
	17. Product innovation and inclusion of healthier options as default choices	1	1	2
	18. Price incentives and marketing strategies used to highlight healthier food and beverage items	0	1	1
	19. Healthy portion sizes are promoted by optimizing the size of dishware and serving ware	0	1	1
	20. Displays, decorations, and signage used to highlight healthier choices	0	4	4
	21. Healthier options promoted via worksite wellness programs or other employee organizations	0	1	1
	22. Changes to vendor contracts that support food service guidelines	0	1	1

* Only guidelines that were present as a policy in at least one contract or reported as a non-contractual practice are listed in this table. ** Guidelines were not written into their food service contracts, but universities implemented them as a non-contractual practice.

## Data Availability

The data presented in this study are available on request from the corresponding author.

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
