# Peer review of "Healthy Food Policies Documented in University Food Service Contracts"

_ijerph, 2023, doi:10.3390/ijerph20166617_

Round 1

Reviewer 1 Report

It’s an interesting and relevant study for public health nutrition that assesses whether healthy food standards and policies were incorporated into the contract agreements between North Carolina (NC) public universities and their food service management companies.

The study seems to be well conducted. My main concerns are related to the necessity to clarify some methodological aspects and results. Additionally, some inclusions about the food environment in the Introduction can contribute to the article.

Lines 32-38. Turner et al framework was adopted in the study, but considering that universities are organizational food environments and that interventions in this setting are different from interventions in the community environment, due to its public, governance, and other elements, some considerations specifically related to organizational food environments should be included. Furthermore, it is important to establish a dialogue between the dimensions and components presented in the model and introduction and the dimensions and components assessed in the study.

Lines 162-167. How data was analyzed? Was there a combination of Policies Identified in University Contracts and Non-Contractual Practices Reported by University Representatives? Or were there considered only policies presented in the contract? It should be cleared.

Lines 169-170. How the information about the length of the contract was collected? It was not mentioned in the Methods.

Lines 170-171. Could the same contract encompass different services? Or were there specific contracts for different services, considering their specificities? It should be cleared.

Lines 172-174. Did these numbers include Non-Contractual Practices Reported by University Representatives?

Line 215. It is important to reinforce that the maximum number of policies and practices adopted by universities was 9 of 35.

What thoughts do the authors have about improving the contracts in the universities? What strategies and actors should be involved? 

Author Response

Response to Reviewer 1 Comments

Point 1: Lines 32-38. Turner et al framework was adopted in the study, but considering that universities are organizational food environments and that interventions in this setting are different from interventions in the community environment, due to its public, governance, and other elements, some considerations specifically related to organizational food environments should be included. Furthermore, it is important to establish a dialogue between the dimensions and components presented in the model and introduction and the dimensions and components assessed in the study.

Response 1: Thank you for suggesting incorporating organizational food environment elements into the manuscript. We agree that explaining and connecting our work to this unique food environment will strengthen the paper. We added a paragraph in the introduction (lines 39-46) and referenced the concept in the discussion (lines 275-277) and conclusion (341-343).

Point 2: Lines 162-167. How data was analyzed? Was there a combination of Policies Identified in University Contracts and Non-Contractual Practices Reported by University Representatives? Or were there considered only policies presented in the contract? It should be cleared.

Response 2: Descriptive statistics were used for this analysis, including the mean and median contract length and count of guidelines as contractual policies and non-contractual practices. We examined policies and non-contractual practices separately, then together for the universities that provided feedback (n=7). We added more information to our analysis section to clarify (lines 180-187).

Point 3: Lines 169-170. How the information about the length of the contract was collected? It was not mentioned in the Methods.

Response 3: We added a sentence about the variables we extracted from each contract, including the contract year. See lines 156-159.

Point 4: Lines 170-171. Could the same contract encompass different services? Or were there specific contracts for different services, considering their specificities? It should be cleared.

Response 4: We added a new Table 1 and expanded the narrative surrounding the settings used to sell/serve food on campus (lines 190-197). Each contract included a variety of settings that the contractor was responsible for; all contracts included dining halls and catered events, and some included additional services such as vending machines and hospital patient meals.

Point 5: Lines 172-174. Did these numbers include Non-Contractual Practices Reported by University Representatives?

Response 5: We changed the language to read ‘food service contracts’ instead of ‘universities’ to clarify that we are referring to policies (i.e., guidelines found in contracts) (lines 202-204).

Point 6: Line 215. It is important to reinforce that the maximum number of policies and practices adopted by universities was 9 of 35/

Response 6: We agree this should be reiterated and have incorporated your suggestion on lines 256-257.

Point 7: What thoughts do the authors have about improving the contracts in the universities? What strategies and actors should be involved?

Response 7: We elaborated on our thoughts about enhancing food service contracts to include more healthy food guidelines in the discussion and conclusion sections. First, we explained the need to promote the CDC-FSG or other nutrition guidelines to those negotiating contracts (lines 267-269). We also suggest that future research should aim to understand more about the feasibility of incorporating guidelines in contract addendums (lines 269-271). Next, we expanded on how universities can keep health at the forefront of negotiations by inviting public health nutritionists, students, and other campus stakeholders to submit feedback about the importance of healthy food offerings as policies on campus (lines 347-350).

Reviewer 2 Report

This paper describes a content analysis of food service contracts within North Carolina’s public university system using the North Carolina Food Service Policy Guidelines Assessment instrument. The instrument was developed by the Community and Clinical Connections for Prevention and Health (CCCPH) unit, a branch of the Chronic Disease and Injury Section in the North Carolina Division of Public Health.  In addition to the content analysis, the authors contacted each of the universities’ food service departments to corroborate the content analysis and provide an opportunity to add information about any unwritten polices or usual practice employed.  The written policy documents were obtained through Freedom of Information Act and unwritten policies by contacting via with the universities’ food service units.  The authors noted a previous collaboration with CCCPH to assess and modify its own university’s food environment.  Although the intention of this paper was admirable, it is unclear how informative were the data collected.  

Missing Data: Although policy documents were available for the 14 of the 16 state universities that used outside vendors for campus food service, only 50% of those (n=7) provided corroboration of their content analysis or information regarding unwritten policies and usual practice.  [Note:  Table 1 presented the content analysis data and the program directors’ report.   In addition to reporting policies that were practice/unwritten, the table added a percentage value.  This seems inappropriate due to the smaller number of programs responding to this part of the study.  Might want to exclude the use of % for the n=7; seems confusing when this is a subsection of the total group of 14.

No Psychometrics: Also, no data were presented on the psychometrics of the NC Food Service Policy Guidelines Assessment instrument.  The instrument is comprised of 35 policy guidelines related to specific food items that fall within four categories: Beverages, Packaged Snacks, Prepared Foods, and Other.  The Partnership for a Healthier America (PHA) guidelines (noted as having guidelines created with leading nutrition, physical activity, and campus wellness experts).  The PHA website indicates that college/university partners commit to meeting 23 of 41 guidelines within three years.  Within the website, information was posted that NCSU (one of the NC state system universities) was the first to have met 23/41.  Was an effort made to compare content analysis from the NCSU campus using both the PHA and the CCCPH instruments to check for consistency?  Could it be possible for NCSU to score high on PHA but not as well on the CCCPH tool?  Authors should better justify the use of the CCCPH instrument (compared to, for example, PHA instrument).  Would be useful to include a copy in Supplemental Information.  Not easily accessible on the website.)

Possible Bias: Also, the authors noted their university, which is part of the NC state system, had previously collaborated with CCCPH to assess and modify their university food environment.  The authors noted they were encouraged to use the CCCPH tool in this research which could possibly contribute to bias to the study.  

Questionable Data Extraction: Other issues affecting the utility of the research is the use of a single research assistant to extract data.  Best practice suggests the use of at least two coders (and a 3rd when there are disagreements between the two). 

Summarization of Findings/Missing Useful Information:

·       Authors noted that 13 of the 35 NC Food Service Policy Guidelines were in at least one contract among the 14 evaluated.  The Summary also noted that the guidelines, “to offer protein foods from plants such as legumes, nuts, seeds, and soy products” was the most frequently cited.  By how many of these programs?  Were any of the guidelines in ALL of the contracts?   Were any of the guidelines in NONE of the contracts?    Which guidelines were in almost all contracts?  Some? None?  Seems like this information could help formulate next steps to improve the policies employed (by seeing the primary gaps). Maybe summarizing the data differently could add greater understanding.     It might be instructive to summarize presence and absence of guidelines using the summary of the content analysis and the self-reported data for the 7 programs.   Also, were there any common factors associated with the 7 universities that did not respond (e.g., smaller size, less financial resources, etc.). 

·       The authors noted that university food service contracts ranged between 7-14 years.  What was the mean/median length?  What policies were associated with age of contract?    Were the oldest contracts those with fewer guidelines addressed?

·       Conclusions suggest that dissemination efforts should “focus on promoting the NC Food Service Policy Guidelines Assessment or other assessment tools…”.  What efforts have been used to date at the CCCPH following development of the assessment to improve the food environments on university campuses?

Author Response

Response to Reviewer 2 Comments

Point 1: Missing Data: Although policy documents were available for the 14 of the 16 state universities that used outside vendors for campus food service, only 50% of those (n=7) provided corroboration of their content analysis or information regarding unwritten policies and usual practice.  [Note:  Table 1 presented the content analysis data and the program directors’ report.   In addition to reporting policies that were practice/unwritten, the table added a percentage value.  This seems inappropriate due to the smaller number of programs responding to this part of the study.  Might want to exclude the use of % for the n=7; seems confusing when this is a subsection of the total group of 14.

Response 1: We greatly appreciate your feedback on this section of our paper. Although there are 16 public universities in North Carolina, only 14 use food service contracts. Since we received 14 of the 14 contracts, we added lines 187-188, clarifying that we received 100% of the requested contracts. Unfortunately, we only received feedback from 50% of university representatives despite contacting them three times. We sought input from universities for two purposes: 1) as a ‘member check’ to verify our findings and 2) to learn more about the potential for healthy food guidelines residing in another document or agreement between a university and food service provider. To clear up confusion surrounding our original Table 1, we removed the ‘non-contractual practices’ column since it was a subsample of the population. We also eliminated the percentages in the table since our sample size is small (N=14) (now Table 3). We also added a new Table 4 highlighting the feedback received regarding non-contractual practices.

Point 2: No Psychometrics: Also, no data were presented on the psychometrics of the NC Food Service Policy Guidelines Assessment instrument.  The instrument is comprised of 35 policy guidelines related to specific food items that fall within four categories: Beverages, Packaged Snacks, Prepared Foods, and Other.  The Partnership for a Healthier America (PHA) guidelines (noted as having guidelines created with leading nutrition, physical activity, and campus wellness experts).  The PHA website indicates that college/university partners commit to meeting 23 of 41 guidelines within three years.  Within the website, information was posted that NCSU (one of the NC state system universities) was the first to have met 23/41.  Was an effort made to compare content analysis from the NCSU campus using both the PHA and the CCCPH instruments to check for consistency?  Could it be possible for NCSU to score high on PHA but not as well on the CCCPH tool?  Authors should better justify the use of the CCCPH instrument (compared to, for example, PHA instrument).  Would be useful to include a copy in Supplemental Information.  Not easily accessible on the website.)

Response 2: Thank you very much for identifying that more information is needed to justify the use of this tool. Because the CDC funded our project via the CCCPH, they wanted us to use their NC Assessment tool to examine all public universities in North Carolina. Our team agreed that this tool was a good choice for reviewing food service contracts for a couple of reasons:

  1. The CDC-FSG is an evidence-based set of guidelines developed by experts from various federal agencies.
  2. The CDC-FSG is intended for use when developing contracts and permits for food service delivery.

We are aware of the PHA guidelines and referenced them in our introduction. We didn’t feel that the PHA guidelines were appropriate for our project because they include physical activity/movement and programming guidelines (i.e., breastfeeding support) in addition to food and nutrition.

NCSU was not included in our project because they self-operate dining services (they do not have a food service contract to review). However, another university included in our project (UNCG) has also partnered with PHA; their food service contract originated in 2009, and they began their work with PHA in 2018. They were one of the universities that provided feedback but only indicated two additional non-contractual practices beyond what was found in their contract. It would be interesting to learn more about their work with PHA and how it aligns with the CDC-FSG in future research.

Point 3: Possible Bias: Also, the authors noted their university, which is part of the NC state system, had previously collaborated with CCCPH to assess and modify their university food environment.  The authors noted they were encouraged to use the CCCPH tool in this research which could possibly contribute to bias to the study.  

Response 3: Thank you for bringing this to our attention. As previously mentioned, the CDC funded our project through CCCPH; therefore, the funder wanted us to use their tool to conduct our assessment. We hadn’t modified our university’s food environment when we did the statewide contract analysis. Additionally, our work with CCCPH has not influenced any food service contract revisions to date. We aspire to influence policy changes within our university and the university system in the future and hope that this paper will provide evidence to facilitate that conversation. To minimize confusion, we eliminated the sentence about other work we did on campus after this project.

Point 4: Questionable Data Extraction: Other issues affecting the utility of the research is the use of a single research assistant to extract data.  Best practice suggests the use of at least two coders (and a 3rd when there are disagreements between the two).  

Response 4: After the first research assistant extracted data (CW), a second research assistant (SF) reviewed each contract in its entirety and validated the findings (as reflected in the author contributions section of the manuscript, lines 365-366). The team discussed any discrepancies, and a 100% agreement was reached. Lines 151-154 were added to clarify the process. 

Point 5a:   Authors noted that 13 of the 35 NC Food Service Policy Guidelines were in at least one contract among the 14 evaluated. “to offer protein foods from plants such as legumes, nuts, seeds, and soy products” was the most frequently cited.  By how many of these programs? 

Response 5a: Eight universities included ‘offer protein foods from plants such as legumes, nuts, seeds, and soy products.’ We added the count to line 222.

Point 5b: The Summary also noted that the guidelines, Were any of the guidelines in ALL of the contracts?   

Response 5b: Lines 213-214 were added to clarify that no guidelines were found in all contracts. This information is also in Table 3.

Point 5c: Were any of the guidelines in NONE of the contracts?    

Response 5c: Yes, 22 guidelines were not found in any contracts. Please refer to table 3 and line 214.

Point 5d: Which guidelines were in almost all contracts?  Some? None?   

Response 5d: Eight universities included ‘offer protein foods from plants such as legumes, nuts, seeds, and soy products,’ making it the most frequently included guideline. Please refer to Table 3 to view the number of policies present in contracts by guideline. We added more details to lines 217-224, specifying the number associated with the most frequently identified guidelines in each category (e.g., Beverage, Prepared Foods, Other).

Point 5e: Seems like this information could help formulate next steps to improve the policies employed (by seeing the primary gaps). Maybe summarizing the data differently could add greater understanding.     It might be instructive to summarize presence and absence of guidelines using the summary of the content analysis and the self-reported data for the 7 programs.   Also, were there any common factors associated with the 7 universities that did not respond (e.g., smaller size, less financial resources, etc.). 

Response 5e: We added a new Table 1, which provides an overview of the settings used for selling/serving food, and a new Table 2, which presents information by university to examine contract year and the number of guidelines present. We also modified Table 3 (the old Table 1), removing the non-contractual practices column and percentages. Regarding the feedback from universities, we added lines 232-234 explaining that there were no apparent differences between universities that did not respond to our request and those that provided feedback. We are hopeful that presenting the data differently will make it easier to understand.

Point 6:  The authors noted that university food service contracts ranged between 7-14 years.  What was the mean/median length?  What policies were associated with age of contract?    Were the oldest contracts those with fewer guidelines addressed?

Response 6: We updated lines 203-204 to include the mean, median, and range of contract length. We also added a new Table 2 and lines 259-263 that discuss the relationship between contract year and the number of guidelines present. There did not appear to be a relationship between the age of the contracts and the number of guidelines present (see lines 204-206 and Table 2). We feel this information was necessary to point out, so we appreciate the questions.

Point 7: Conclusions suggest that dissemination efforts should “focus on promoting the NC Food Service Policy Guidelines Assessment or other assessment tools…”.  What efforts have been used to date at the CCCPH following development of the assessment to improve the food environments on university campuses?

Response 7: We are unaware of everything that CCCPH is doing with this tool; however, we do know that they received a grant from CDC focused on physical activity and nutrition. This grant was used to support our work.